

# Insight into plant cell wall degradation and pathogenesis of *Ganoderma boninense* via comparative genome analysis

Ahmad Bazli Ramzi, Muhammad Lutfi Che Me, Ummul Syafiqah Ruslan, Syarul Nataqain Baharum and Nor Azlan Nor Muhammad

Institute of Systems Biology, Universiti Kebangsaan Malaysia, Bangi, Selangor, Malaysia

## ABSTRACT

**Background**. *G. boninense* is a hemibiotrophic fungus that infects oil palms (*Elaeis guineensis* Jacq.) causing basal stem rot (BSR) disease and consequent massive economic losses to the oil palm industry. The pathogenicity of this white-rot fungus has been associated with cell wall degrading enzymes (CWDEs) released during saprophytic and necrotrophic stage of infection of the oil palm host. However, there is a lack of information available on the essentiality of CWDEs in wood-decaying process and pathogenesis of this oil palm pathogen especially at molecular and genome levels.

**Methods**. In this study, comparative genome analysis was carried out using the *G. boninense* NJ3 genome to identify and characterize carbohydrate-active enzyme (CAZymes) including CWDE in the fungal genome. Augustus pipeline was employed for gene identification in *G. boninense* NJ3 and the produced protein sequences were analyzed via dbCAN pipeline and PhiBase 4.5 database annotation for CAZymes and plant-host interaction (PHI) gene analysis, respectively. Comparison of CAZymes from *G. boninense* NJ3 was made against *G. lucidum*, a well-studied model *Ganoderma* sp. and five selected pathogenic fungi for CAZymes characterization. Functional annotation of PHI genes was carried out using Web Gene Ontology Annotation Plot (WEGO) and was used for selecting candidate PHI genes related to cell wall degradation of *G. boninense* NJ3.

**Results**. *G. boninense* was enriched with CAZymes and CWDEs in a similar fashion to *G. lucidum* that corroborate with the lignocellulolytic abilities of both closely-related fungal strains. The role of polysaccharide and cell wall degrading enzymes in the hemibiotrophic mode of infection of *G. boninense* was investigated by analyzing the fungal CAZymes with necrotrophic *Armillaria solidipes*, *A. mellea*, biotrophic *Ustilago maydis*, *Melampsora larici-populina* and hemibiotrophic *Moniliophthora perniciosa*. Profiles of the selected pathogenic fungi demonstrated that necrotizing pathogens including *G. boninense* NJ3 exhibited an extensive set of CAZymes as compared to the more CAZymes-limited biotrophic pathogens. Following PHI analysis, several candidate genes including polygalacturonase, endo β-1,3-xylanase, β-glucanase and laccase were identified as potential CWDEs that contribute to the plant host interaction and pathogenesis.

**Discussion**. This study employed bioinformatics tools for providing a greater understanding of the biological mechanisms underlying the production of CAZymes in *G. boninense* NJ3. Identification and profiling of the fungal polysaccharide- and

Corresponding author
Ahmad Bazli Ramzi,
bazliramzi@ukm.edu.my

lignocellulosic-degrading enzymes would further facilitate in elucidating the infection mechanisms through the production of CWDEs by *G. boninense*. Identification of CAZymes and CWDE-related PHI genes in *G. boninense* would serve as the basis for functional studies of genes associated with the fungal virulence and pathogenicity using systems biology and genetic engineering approaches.

## INTRODUCTION

Cell wall degrading enzymes (CWDEs) are part of the carbohydrate-active enzymes (CAZymes) produced by plant pathogens for penetrating and degrading the plant cell walls, and these CAZymes have been directly linked to devastating crop diseases (*Zhang, Bruton & Biles, 2014*; *Somai-Jemmali et al., 2017*; *Gawade et al., 2017*). Plant pathogenic fungi, especially among the fungal families of Ascomycota, Basidiomycota, Chytridiomycota, and Zygomycota, have been reported to contain the highest number of CAZymes (*Zhao et al., 2013*; *Kubicek, Starr & Glass, 2014*). Differences in composition and structure of the woody components are commonly mirrored with the types of lignocellulolytic enzymes produced by invading pathogenic fungi (*King et al., 2011*). In fact, many plant pathogens particularly white rot fungi are well-endowed with high copies of CWDEs as compared to decay-feeding saprotrophs and have been demonstrated to be highly competent producers of lignocellulolytic enzymes for host-specific attack, and subsequent biomass degradation (*King et al., 2011*; *O'Connell et al., 2012*).

Interestingly, genome and transcriptome analyses indicate different profiles of CAZymes were produced by pathogenic fungi during different stages of pathogenesis transition from biotrophic to necrotrophic lifestyles (*O'Connell et al., 2012*; *M'Barek et al., 2015*). Enzymatic production of CWDE by pathogenic fungi has been found to be correlated with the degree of pathogenicity and cell wall disintegration of infected plant hosts (*Kang & Buchenauer, 2000*; *Wanjiru, Zhensheng & Buchenauer, 2002*; *Lyu et al., 2015*; *Somai-Jemmali et al., 2017*). Considering the importance of CWDE in the uptake of nutrients from plant host, these hydrolytic enzymes are considered as the key pathogenicity determinant among plant pathogens (*Brito, Espino & González, 2006*; *Kubicek, Starr & Glass, 2014*; *Bravo Ruiz, Di Pietro & Roncero, 2016*).

*Ganoderma boninense* is a causative agent of basal stem rot (BSR) disease that beset the oil palm industries with devastating economic losses due to the reduced lifespan and eventual death of the infected oil palm (*Chen et al., 2017*). Due to the toxicity and environmental issues of chemical pesticides, the *Ganoderma* disease is currently managed mainly through cultural practices such as the removal of dead trees and infected stumps prior to or during replanting but these strategies remain ineffective in preventing the spread of the *G. boninense* in affected plantations (*Hushiarian, Yusof & Dutse, 2013*; *Sahebi et al., 2015*). Recent research works in overcoming the *Ganoderma* disease have been

mainly aimed at understanding the oil palm molecular defense response via transcriptional analysis, and profiling of proteins and metabolites of the infected oil palm (*Nusaibah et al., 2016*; *Sahebi et al., 2017*; *Ho et al., 2018*). The spread of *G. boninense* in oil palm plantation has been attributed to two main routes specifically spore dispersal and root contact with *G. boninense* infected palm tissues (trunk, bole, and roots) (*Paterson, 2007*; *Chen et al., 2017*). Importantly, root infection via cell wall degradation has been suggested as the main mode of *Ganoderma* infection based on the spread of infection from root to the base of mature palm trees (*Rees et al., 2009*).

Lignocellulolytic enzymes of *G. boninense* have been shown to be predominant in instigating oil palm infection and cell wall-degrading processes (*Goh, Ganeson & Supramaniam, 2014*; *Jumali & Ismail, 2017*; *Surendran et al., 2018*). Direct roles of CWDEs in the hemibiotrophic infection of oil palm roots were first demonstrated via macroscopic examination of enzymatically-degraded root outer cell layers and invaded root and stem tissues of *G. boninense*-infected tissues (*Rees, 2006*; *Rees et al., 2009*). In the initial stage of infection, the fungal mycelia behave as biotrophs that absorb the plant nutrient by penetrating the oil palm root surface and culminating in rapid growth spread in oil palm lower stem. During the necrotrophic stage, the fungus attacks the host cell walls by excreting host of enzymes including CWDEs that led subsequent cell death and multiplication of basidiocarps in decayed palm woods (*Rees et al., 2009*; *Chong, Dayou & Alexander, 2017*).

Despite the important roles of CWDEs in the oil palm pathogenesis, information about genomic features and mechanisms underlying the pathogenicity of *G. boninense* in oil palm is severely lacking. The necessity for establishing a reliable genetic model for understanding *Ganoderma*-oil palm interactions is highlighted in recent reports of draft genome sequences of different strains of *G. boninense* which could facilitate the identification of CWDEs as pathogenicity factors essential for successful invasion of oil palm cells (*Sulaiman et al., 2018*; *Utomo et al., 2018*). A deeper understanding on the genetic composition of CWDEs as part of carbohydrate-acting enzymes of *G. boninense* and the biological mechanisms conferring the fungal ability to produce CWDEs are important in our efforts to elucidate the plant-pathogen interactions at the genome and molecular levels. Therefore, this research work was devised to obtain genomic insight of CWDEs in *G. boninense* through computational and comparative genome analysis. Genome sequence of *G. boninense* NJ3 that was isolated from Indonesian oil palm field was used for the comparative genome analysis (*Mercière et al., 2015*). In this study, CAZymes specifically auxiliary protein (AA), glycosyltransferase (GT), carbohydrate binding modules (CBMs), carbohydrate esterases (CE), glycoside hydrolases (GH) and polysaccharide lyases (PL) in *G. boninense* NJ3 genome were annotated using CAZy annotation pipeline. Direct comparison of CAZymes was made with close relative and model strain *G. lucidum* as the reference *Ganoderma* strain. Further comparison was carried out with five selected pathogenic Basidiomycetes in the search for genetic patterns underlying *G. boninense* hemibiotrophic infection strategy. Identification of the responsible genes for *G. boninense* CAZymes including CWDEs will broaden genomic understanding on the molecular mechanisms of the fungal wood-decaying abilities and oil palm pathogenesis.

## MATERIALS & METHODS

### *Ganoderma boninense* NJ3 genome

The *G. boninense* NJ3 assembled genome was obtained from the NCBI assembly database (GenBank assembly accession: GCA_001855635.1). The assembly level is at contig level and was sequenced using Illumina HiSeq 454 in 2016. Comparative genome analysis was carried out by comparing CAZymes dataset of *G. boninense* NJ3 with previously-reported genome sequence of *Ganoderma lucidum* (*Chen et al., 2012b*), *Melampsora larici-populina* (*Duplessis et al., 2011*), *Ustilago maydis* (*Kämper et al., 2006*), *Moniliophthora perniciosa* FA553 (*Mondego et al., 2008*), *Armillaria solidipes* 28-4 (*Sipos et al., 2017*) and *Armillaria mellea* DSM 3731 (*Collins et al., 2013*). Except for *G. lucidum*, archival CAZymes information of the other referenced fungal genomes was retrieved from Joint Genome Institute MycoCosm portal in May 2018 (*Grigoriev et al., 2014*) (https://mycocosm.jgi.doe.gov/mycocosm/home).

### Prediction of genes

The contigs from the *G. boninense* NJ3 assembly was processed using Augustus gene prediction tool for the identification of genes (http://augustus.gobics.de/). Gene prediction was carried out using ''augustus –species=phanerochaete_chrysosporium gboninense_NJ3.fna >gboninense_NJ3_augustus.gff'' command where gboninense_NJ3.fna was the assembled contigs file. *Phanerochaete chrysosporium* was chosen as the gene prediction model as it was the closest species to *G. boninense* in Augustus based on the NCBI taxonomy browser (https://www.ncbi.nlm.nih.gov/Taxonomy/Browser/wwwtax.cgi). A GFF file containing the genes predicted and their annotations was produced by Augustus and the protein sequences were then extracted via the command ''getAnnoFasta.pl gboninense_NJ3_augustus.gff''. The predicted genes dataset of the *G. boninense* NJ3 genome has been deposited at European Nuclear Archive (ENA) under the accession number PRJEB34805.

### dbCAN pipeline analysis

The produced protein sequences from Augustus was searched against the dbCAN: An HMM (Hidden Markov Model) based database for carbohydrate-active enzyme annotation. dbCAN release 6.0 was downloaded in May 2018. The downloaded database was converted into a HMM formatted database using hmmpress (part of HMMER3 software package). hmmscan was run with the following parameters; –domtblout results.out.dm. hmmscan-parser.sh (from dbCAN) was used to process the results table with e-value 1E-3 as filter.

### Annotation by PhiBase 4.5 database

PhiBase 4.5 was downloaded for local analysis from http://www.phi-base.org. The raw sequence file was converted into a blast database using makeblastdb (part of the ncbi-blast+ software package). A local blastp run was deployed to identify homologs of PhiBase 4.5 in *G. boninense* NJ3 predicted protein sequences. Blastp was run with the parameters -outfmt 6 -max_target_seqs 1 -max_hsps 1 -evalue 0.1.

## Gene ontology and WEGO chart

Gene ontology of PhiBase 4.5 homologs in *G. boninense* NJ3 was obtained using Blast2GO 5.2.5 pipeline. Using local blastp function in Blast2GO, the sequences of the homologs were annotated against the NCBI NR (non-redundant) protein database. The NR database was downloaded in April 2019. *E*-value chosen was 0.1 and number of blast hits was set to 10. Other parameters remained at default values. The gene ontologies were then mapped onto the sequences by mapping the latest Blast2GO database with the blasted sequences. Next, InterProScan 5.33–72 was ran locally to obtain protein domain annotations. The XML file produced was then loaded in Blast2GO. Lastly, the annotation tool in Blast2GO merged and verified the gene ontologies obtained between both gene ontology annotation methods. Blast2GO annotation parameters were left at default values. Functional classification of PHI genes was carried out using Web Gene Ontology Annotation Plot (WEGO) software (*Ye et al., 2006*). To generate the WEGO chart, the results from Blast2GO annotation were exported in WEGO native format. The WEGO chart was then generated by uploading to http://wego.genomics.org.cn/. Datasets of *G. boninense* NJ3 Augustus gene annotation, GO annotation and protein ID from PhiBase 4.5 analysis are provided in supplementary files.

# RESULTS & DISCUSSION

## Characterization of carbohydrate-active enzymes (CAZymes) in *G. boninense*

In this study, the draft genome of *G. boninense* NJ3 was used for identifying and characterizing the carbohydrate-active enzymes in fungal genome. To identify CAZymes in *G. boninense* NJ3, the assembled dataset was further processed using Augustus pipeline to produce predicted genes peptide sequence for CAZymes analysis via the dbCAN pipeline. Comparative analysis for CAZymes characterization in *G. boninense* was carried out using *G. lucidum* genome sequence as the reference *Ganoderma* spp. strain owing to the high-quality genome sequence and well-established genomics studies of this closely-related *Ganoderma* strain. We hypothesized that *G. boninense* is enriched with a high number of CAZymes similar to *G. lucidum* that provide wood-degrading capabilities and contribute to the disparate nutritional strategy of hemibiotrophic *G. boninense* and saprophytic *G. lucidum*, respectively.

Following analysis, a total of 755 CAZymes was identified in *G. boninense* NJ3 as compared to 489 CAZymes found in *G. lucidum* (Fig. 1). Overall, about 465 copies of cell wall degrading enzymes (CWDEs) comprising of glycoside hydrolase (GH), carbohydrate esterase (CE) and polysaccharide lyase (PL) were found in the *G. boninense* NJ3 genome. From the total CWDEs, 348, 102 and 15 genes were found for GH, CE and PL, respectively (Table S1). The amount of CWDEs found in *G. boninense* is comparatively higher when compared with *G. lucidum* (273 GHs, 30 CEs, 10 PLs). A richer and highly similar set of CWDEs was observed in *G. boninense* NJ3 that enables degradation of woody structures such as hemicellulose and pectin for nutrient uptake and growth in similar means to *G. lucidum* which harbored one of the richest sets of polysaccharide-degrading enzymes in the sequenced genome of Basidiomycota fungi (*Chen et al., 2012b*). Interestingly, *G.*

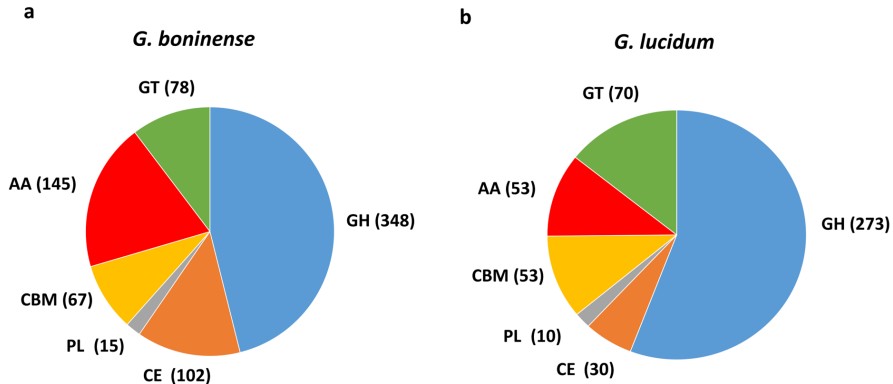

**Figure 1** **Comparison of carbohydrate-active enzymes (CAZymes) in *Ganoderma* spp.** Overview of CAZymes profile in (A) *G. boninense* NJ3 and (B) *G. lucidum*. Abbreviations: GH Glycoside hydrolase, CE Carbohydrate esterase, PL polysaccharide lyase, CBM carbohydrate binding module, AA auxiliary activity, GT glycosyltransferase.

*boninense* NJ3 possesses a higher number of CEs of about 102 copies as compared to *G. lucidum* with 30 copies of CEs that are important for plant cell wall modification. In addition to polysaccharide-deacetylating CE enzymes, *G. boninense* NJ3 also harbors a diverse array of GHs that are crucial in the hydrolysis of cellulose and hemicellulose components of the plant biomass. Common GH in white rot fungi such as GH6 and GH7, and universal CWDEs of cellulolytic GH1, GH3 and GH5 for degradation of cellulose, hemicellulose and pectin were observed in both *Ganoderma* spp. Importantly, *G. boninense* possesses several GHs that were not found in *G. lucidum* specifically polysaccharides-acting GH109 (α-N-acetylgalactosaminidase), GH145 (L-Rh α-α-1,4-GlcA α-L-rhamnohydrolase), GH135 (α-1,3-galactosaminogalactan hydrolase) and GH131 (broad specific β-glucanase) that degrade both cellulose and hemicellulose. In terms of polysaccharide-active enzymes, both *Ganoderma* spp. contained multiple copies of pectic-acting PL8 and PL14 while PL4 (rhamnogalacturonan endolyase), PL12 (heparin-sulfate lyase) and PL15 (alginate lyase) were only found in *G. boninense*.

Apart from CWDEs, both fungal genomes harbored other CAZymes including carbohydrate binding module (CBM), auxiliary activity (AA) and glycosyltransferase (GT) that are essential for lignin depolymerization and carbohydrate utilization from the host plant. During wood decaying process, access to the structural woody components was aided by CBMs that formed a two-domain structure together with catalytic domains (CDs) of cellulases by increasing the enzyme concentration on the substrate surfaces. Overall, a total of 290 copies of CAZymes was identified in *G. boninense* NJ3 as compared to 176 copies in *G. lucidum*. Of these, 67 CBMs, 145 AAs and 78 GTs were identified from *G. boninense*. Both *Ganoderma* spp. strains have a similar set of CBMs except for CBM19 (chitin-binding function) and CBM32 (pectic-binding) were unique for *G. boninense* while CBM12 (chitin-binding) was only found in *G. lucidum*. Although white rot fungi have been

associated with the lack of CBMs, *G. lucidum* and *G. boninense* contained 10 and 12 out of 16 total families of CBMs, respectively. It is notable that *G. boninense* possessed high copies of CBM1, an important fungal CBM that uses cellulose and chitin as substrates for polysaccharide-degrading activities (*Mello & Polikarpov, 2014*; *Várnai et al., 2014*). Both fungi shared similar GTs except for GT65 and GT41 that were found only in *G. boninense* while GT31 was present only in *G. lucidum*. The ability of *Ganoderma* spp. to utilize nutrient from plant tissues relied heavily on the synergistic actions of cellulolytic and ligninolytic enzymes that include redox AA enzymes (*Zhou et al., 2018*). Laccase (AA1_A1), ferroxidase (AA1_A2), class II peroxidase (AA2), GMC oxidoreductase (AA3), radical-copper oxidase (AA5), 1,4-benzoquinone reductase (AA6) and iron reductase (AA8) were among AA enzymes identified in both sets of fungal genomes. From this comparative analysis, *G. boninense* was found to be endowed with significantly higher copies of the redox enzymes especially lignin-acting laccase (AA1) and peroxidase (AA2) and oxidoreductase (AA3) enzymes as compared to *G. lucidum*.

Lignocellulolytic enzymes production has been well-documented in white rot *Ganoderma* spp. for wood decomposition and subsequent feeding and propagation on the woody substrates (*Silva, Melo & Oliveira, 2005*; *Paterson, 2007*; *Zhou et al., 2013*). Apart from wood-degrading enzyme producing capabilities and plant pathogenicity, *Ganoderma* species are generating much research interest for therapeutic applications through the production of bioactive polysaccharides and terpenoids such as ganoderic acid (*Boh, 2013*; *Wu et al., 2013*). Owing to its therapeutic and biotechnological potentials, *G. lucidum* has been developed as model medicinal mushroom through extensive biochemistry, genomics, and genetic engineering research, and this saprophytic mushroom is in fact endowed with an extensive set of CAZymes encoded in its genome (*Xu, Xu & Zhong, 2012*; *Chen et al., 2012b*; *Liu et al., 2012*; *Yu et al., 2012*). In this study, the genome sequence of *G. boninense* NJ3, a pathogenic fungal isolate from oil palm plantation in Indonesia (*Mercière et al., 2015*), was employed for identifying genes involved in the production of cell wall degrading and carbohydrate active enzymes. By comparing CAZymes of *G. boninense* NJ3 with model *G. lucidum*, profiles of these closely-related *Ganoderma* spp. can be acquired especially in terms of the cell wall degrading abilities of the lesser-studied *G. boninense*. Based on the results obtained, *G. boninense* NJ3 was found to be enriched with an extensive repertoire of CAZymes in similar fashions but with significantly higher numbers of lignocellulosic-degrading enzymes as compared to the non-pathogenic *G. lucidum*, hence, underlining the essentiality of CAZymes in cell wall degradation for the fungal growth and nutrient uptake. Differences in CAZymes characteristics especially CWDEs and polysaccharide-active AAs can be linked and predetermined by the nutritional strategy of either *Ganoderma* spp. hence providing genomic insight and characterization of plant cell wall degradation mechanism of these industrially-important fungi.

## Profiling of CAZymes in selected phytopathogenic fungi

Following characterization of CAZymes in *G. boninense*, the innate ability of this white rot fungus to cause oil palm BSR disease was further investigated by comparative analysis with a few selected disease-causing Basidiomycetes. For this purpose, five phytopathogenic

basidiomycetous fungi exhibiting the biotrophic, hemibiotrophic and necrotrophic mode of plant infections were employed for comparison with *G. boninense*. The fungi of interest were *Ustilago maydis* (model biotrophic pathogen), *Melampsora larici-populina* (biotrophic poplar pathogen), and *Moniliophthora perniciosa* (hemibiotrophic cacao pathogen). The remaining two fungi were *Armillaria solidipes* and *A. mellea* representing facultative necrotrophic fungi attributed to root rot in many conifers and ornamentals, respectively (*Kämper et al., 2006*; *Duplessis et al., 2009*; *Meinhardt et al., 2014*; *Koch et al., 2017*). Biotrophs primarily depend on and derive nutrients without killing the hosts while necrotrophs kill the plant and feed nutrients off the dead cells (*Mendgen & Hahn, 2002*). On the other hand, hemibiotrophs adopt early asymptomatic biotrophic phase and then switched to the host-killing necrotrophic stage with distinct disease symptoms, and decayed tissues (*Horbach et al., 2011*). Although each fungus may differ in targeting host and infection mechanisms, these plant pathogens have been found to rely on an array of hydrolytic enzymes for complete degradation of plant biomass for colonization and nutrient uptake with or without killing the hosts. In this study, we hypothesized that pathogenic fungi with necrotizing abilities (necrotroph and hemibiotroph) would harbor distinct CAZymes profile as compared to non-necrotizing fungi (biotroph) which may be attributed to specific host preference and interaction.

The profile of CWDE in all six pathogenic fungi was illustrated in Fig. 2. *G. boninense* NJ3 contained glycosyl hydrolase (GH) GH2 and GH10 for specialized hemicellulose degradation in addition to dual cellulose and hemicellulose-degrading activities of GH1, GH3, GH5, GH12, GH51, and GH131. The ability of this oil palm pathogen to hydrolyse the pectin component is further provided by GH28, GH105 and necrotroph-specific GH53 and GH78. Biotrophic *U. maydis* and *M. laricis-populina* lack GH1, GH6, GH78 and GH95 which were prevalent in the necrotizing fungi (*G. boninense* NJ3, *M. perniciosa*, *A. solidipes* and *A. mellea*) (Fig. 2A). Additionally, *U. maydis* lacked GH7 that is common among pathogenic white rot fungi. On the other hand, biotrophic *U. maydis* and *M. laricis-populina* possess GH26 which was not observed in other 4 necrotizing pathogens investigated in this study. The lack of GH1, GH6 and GH78 were well-documented in biotrophs which generally harbor less plant cell wall degrading enzymes than necrotrophs and hemibiotrophs (*Zhao et al., 2013*; *Li et al., 2017*). Obligate necrotrophs (*A. solidipes* and *A. mellea*) and hemibiotrophs (*G. boninense* NJ3 and *M. laricis-populina*) were evidently supplemented with GH3 and GH28 for cellulose, hemicellulose and pectin degradation. From the analysis, *G. boninense* NJ3 exhibited the highest copies of GH18 (Chitinase/endo-β-N-acetylglucosaminidase), GH43 (Hemicellulase), GH79 (Glucuronidase), GH10 (Xylanase) and harbored unique GH4 (glycosidase), GH89 (α-N-acetylglucosaminidase) and GH109 (α-N-acetylgalactosaminidase) for polysaccharide depolymerization in comparison to other plant pathogens examined in this study. Taken together, these findings highlighted the essentiality of several GHs specifically GH3 and GH5 for cell wall degradation by the phytopathogens that corroborated with previous reports on the plant host infection interplays by the phytopathogenic fungi (*Zhao et al., 2013*; *Blackman, Cullerne & Hardham, 2014*; *Chang et al., 2016*).
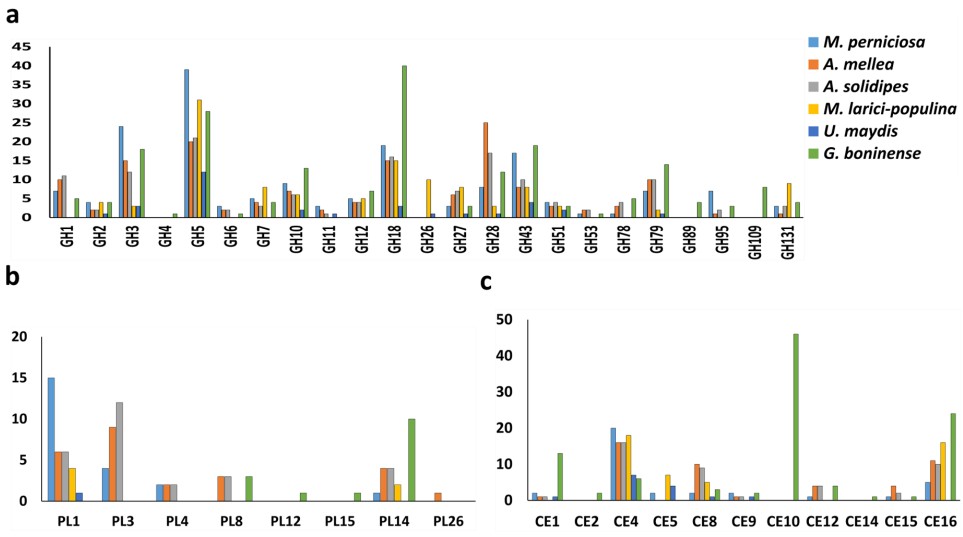

**Figure 2 Profiling of CWDEs in selected phytopathogenic Basidiomycota fungi.** Profile of (A) common GH, (B) CE and (C) PL in the pathogenic fungi. Abbreviations: GH Glycoside hydrolase, CE Carbohydrate esterase, PL polysaccharide lyase.

In addition to cellulose and hemicellulose, some plants are enriched with pectins comprising of homogalacturonan, xylogalacturonan or rhamnogalacturonan as external barriers against pathogen infections. In *G. boninense* NJ3, the cell wall-degrading GHs could work in tandem with pectic-acting enzymes of polysaccharide lyase 8 (PL8), PL12, PL14 and PL15. Common PL found in pathogens, PL1 and PL3 were not observed in *G. boninense* NJ3 which is interesting considering the abundance of these PLs in necrosis-causing *M. perniciosa, A. solidipes* and *A. mellea* (Fig. 2B). Pectin degradation by pectinolytic enzymes particularly PL4 is common among necrotizing fungi examined in this study except *G. boninense* NJ3 and this enzyme had been shown to be highly expressed during crops infection by necrotrophic *Rhizoctonia solani* (*Zheng et al., 2013*; *Chang et al., 2016*). Although important for cell wall degradation by fungi, the smaller amount of pectinases in *G. boninense* NJ3 may indicate substrate or host preference specifically monocotyledon-type as compared to dicotyledon-preferred pathogens that have been associated with increased secretion of pectinases (*Zhao et al., 2013*; *Loyd et al., 2018*). Hemibiotrophic and necrotrophic fungi are well-equipped with the extended set of CWDEs which enable tailored and extensive production of the cell wall degrading enzymes during infection.

In this study, it was found that all 6 pathogenic fungi possess at least one copy of carbohydrate esterase 4 (CE4) as one of the polysaccharide-modifying enzymes in the genomes (Fig. 2C). The genome of *G. boninense* NJ3 was well-represented with CE16 in addition to CE1 and CE12 that were also found in necrosis-causing *M. perniciosa, A. solidipes* and *A. mellea* while CE2, CE14 and high copies of CE10 were found only in the pathogenic *Ganoderma* spp. CEs have been associated in the first line of attack during fungal invasion via the removal of acetylated moieties of saccharides that formed parts of plant

protection system against hydrolytic enzymes (*Ospina-Giraldo, McWalters & Seyer, 2010*; *Sista Kameshwar & Qin, 2018*). The CE10 enzyme is involved in the degradation of lignin and cellulosic components of the plant cell wall and was found to be abundant in several pathogenic fungi including *Macrophomina phaseolina*, *Bipolaris cookei* and *Corynespora cassiicola* (*Islam et al., 2012*; *Zaccaron & Bluhm, 2017*; *Looi et al., 2017*). In sum, notable differences in CWDE profiles of hemibiotrophic and necrotrophic fungi can be associated with the less aggressive nature of biotrophic *U. maydis* and *M. laricis-populina* that adapted the hydrolytic enzyme production specifically for limiting host cell wall damages hence supporting their host nutrient-dependent growth (*Kämper et al., 2006*; *Duplessis et al., 2009*; *Olson et al., 2012*).

Profiling of the remaining CAZymes in the six pathogenic fungi was carried out for comparing and establishing the association between mode of infection and type of genes present. Generally, glycoside transferase (GT) enzymes are mainly responsible for cell wall formation in contrast to the more abundant carbohydrate-hydrolysing GHs in the fungal genomes. As shown in Fig. 3A, the six pathogenic fungi harbored highly similar set of GTs while GT71 is unique for biotrophs and GT65 was only found in *G. boninense* NJ3. Metabolism of starch components of the plant biomass is linked to the presence of starch-active carbohydrate binding modules (CBMs) including CBM1, CBM20, CBM48 and CBM50. Only CBM48 and CBM50 were found in all of the studied pathogens while CBM1 was missing in biotrophic *U. maydis* and *M. laricis-populina* which in turn, represented the most in *G. boninense* NJ3 genome (Fig. 3B). The occurrence of CBMs was often associated with facilitating the hydrolytic activities of amylolytic GHs such as GH13 and GH15 by increasing cell-substrate attachment and degradation (*Chen et al., 2012a*). In particular, CBM1-containing proteins have been found mainly in necrotrophs and hemibiotrophs for promoting cellulose hydrolysis and were shown to elicit plant defense response which is detrimental for fungi with biotrophic lifestyle (*Jones & Ospina-Giraldo, 2011*; *Klosterman et al., 2011*; *Larroque et al., 2012*).

For lignin decomposition, all six pathogenic fungi possessed auxiliary activities (AA) enzymes for AA1, AA3, AA5 and AA6 that encode for ligninolytic and redox enzymatic activities while phenolic-active AA4 (vanillyl-alcohol oxidase) was found only in *G. boninense* NJ3 (Fig. 3C). Another important lignin-modifying enzyme, AA9 also classified as lytic polysaccharide monooxygenases (LPMO), is involved in lignocellulosic degradation by oxidizing cellulose in synergistic reactions with laccase and lignin-modifying peroxidase enzymes. On one hand, biotrophic fungi *M. larici-populina* and *U. maydis* harbored a smaller set of lignolytic AAs with about 36 (*M. larici-populina*) and 23 (*U. maydis*) as compared to the other studied pathogens. These biotrophic strains contained lesser copies of AA1 encoding for laccase and multicopper oxidase enzymes that involved in degradation of lignin barrier. All pathogens possessed AA9 except *U. maydis* while AA2 and AA8 were absent in both biotrophic fungi. High copies of AA9 observed in necrotrophs and hemibiotrophs studied may indicate the importance of these enzymes during the host attack and cell wall deformation. The identification of cellulose-cleaving oxidoreductases LPMOs as part of that AA9 family was previously associated with improved fungal cellulase and wood decaying activities with the presence of reducing

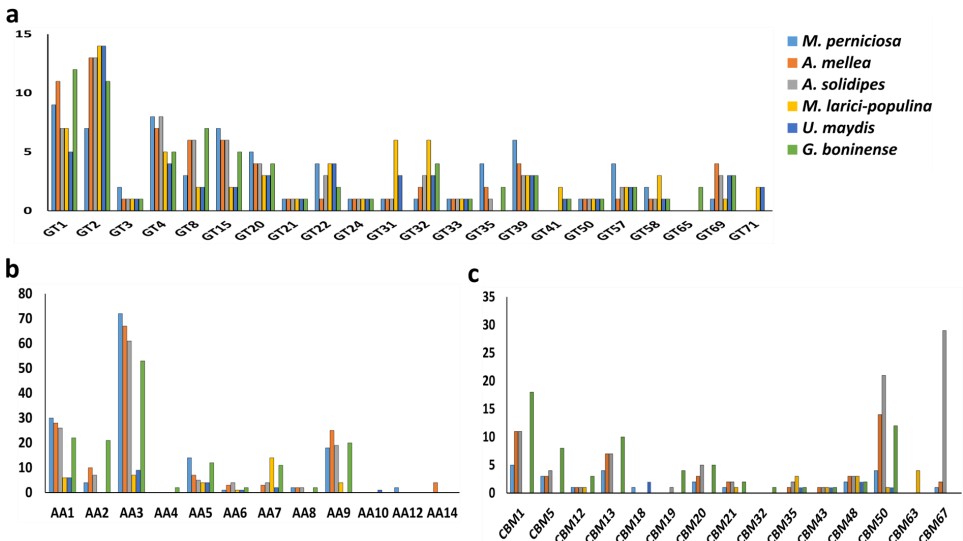

**Figure 3** **Profiles of carbohydrate-active enzymes (CAZymes) in phytopathogenic Basidiomycota fungi.** Profiles of (A) selected GT, (B) AA and (C) CBM of the compared plant pathogens. Abbreviations: GT, glycosyltransferase; AA, auxiliary activity; CBM, carbohydrate binding module.

agents (*Dimarogona, Topakas & Christakopoulos, 2013*; *Karnaouri et al., 2014*). These auxiliary redox enzymes played an important role in completing hydrolysis of lignin by wood-decomposing saprotrophic fungi and have been associated with increased virulence of parasitic fungi (*Hatakka, 1994*; *Levasseur et al., 2013*; *Janusz et al., 2017*). The abundance of AAs may contribute to the enhanced ability of *G. boninense* NJ3 to invade and penetrate lignin and acetylated saccharides as it switches from biotrophic to necrotrophic parasitism that involves overlapped biological processes as found in forest pathogen and wood decayer *Heterobasidion annosum sensu lato* (*Olson et al., 2012*). The production of diverse ligninolytic enzymes by *Ganoderma* are therefore important for the fungal proliferation off plant tissues especially in the depolymerizing of the recalcitrant lignin barrier (*Hu et al., 2017*; *Sarah Jumali & Ismail, 2017*; *Zhou et al., 2018*).

Expression patterns and production of carbohydrate-acting enzymes have been demonstrated to be correlated with the fungal mode of interactions with host plants. Transcriptome analysis of *G. boninense*-treated oil palm transcripts showed very high expression of a host of distinct up-regulated genes encoding for CAZymes from lignin-degrading AAs (laccase and AA2 manganese peroxidase), carbohydrate-active CBM and CE (CBM13, CE10, CE9) to cell wall-hydrolyzing exo-β-1,3-glucanase, chitinase and polygalacturonase when compared to untreated and *Trichoderma harzianum*-treated control samples (*Ho et al., 2016*). Similar patterns of highly expressed CAZymes transcripts were observed in necrotrophic *A. solidipes* that exhibited high number of homologs of GH18, GH47, CE10, CE4 and polygalacturonase following plant-fungus inoculation (*Ross-Davis et al., 2013*). Higher expression of cell wall degrading enzymes (GH, PL, GT) were observed in necrotrophic *Leptosphaeria biglobosa* as compared to hemibiotrophic
counterpart, *L. maculans* which accumulated higher CBM during early stage of plant infection (*Lowe et al., 2014*). Similar CAZymes interplays were suggested in the early infection of *G. boninense* that aimed at overcoming the oil palm host defense response mechanisms including hypersensitive response (HR) leading to the switch from biotrophic stage to the more aggressive necrotrophic attacks culminating in host cell death and successful invasion (*Bahari et al., 2018*). A closer look of the CAZymes in the selected pathogenic fungi would therefore enable genome-wide profiling of carbohydrate-active enzymes that are distinct and correlated with the fungal mode of infection. Importantly, *G. boninense* NJ3 harbored a distinct set of cell wall degrading and polysaccharide depolymerization enzymes that were suited for infecting monocot oil palm host through hemibiotrophic lifestyle.

## Potential pathogenicity genes among CAZymes of *G. boninense* NJ3

Comparative CAZymes analysis of the selected of phytopathogens indicated the correlation between the fungal nutritional strategy with the profiles of carbohydrate-active enzymes essential for plant host cell wall degradation and nutrient consumption. Considering the lack of information of the genes related to the pathogenicity of *G. boninense*, further genome-wide analysis of the fungal genome was carried out using the protein sequences in the Pathogen-Host Interaction Database (PHI database) and functionally classified according to molecular function, biological process and cellular component via WEGO analysis. A total of 5,099 annotated PHI genes were obtained from the WEGO analysis of which membrane (1,682, 24.8%) and metabolic process (2,903, 42.8%) were represented the highest in respective cellular component and biological process categories (Fig. 4). In molecular function category, the PHI genes were predominantly annotated with catalytic activity (3337, 49.2%) including CAZymes-related polygalacturonase (GO:0004650), cellulase (GO:0008810) and endo-1,4-beta-xylanase (GO:0031176).

Considering the prevalence of carbohydrate-active enzymes and the high percentage of PHI genes with hydrolase activity, we hypothesized that some of the CAZymes may directly involve in plant pathogenesis via cell wall degradation by the secreted enzymes. As shown in Table 1, several genes of *G. boninense* NJ3 were shown to share PHI homologs with lignin depolymerization and cell wall degrading enzymes specifically pectic-acting polygalacturonase (PG)-coding homolog gene (PHI id:4879), endo-1,4-beta-xylanase GH10 (PHI id:2209), β-glucanase Eng1 (PHI id:6265) and laccase LCC2 (PHI id:552). CWDEs including pectinase, glycosyl hydrolase and laccase mainly serve as primary weaponry for fungal attack causing the plant cell wall becoming less compact and more permeable for consequent digestion by cellulase and hemicellulase enzymes (*Chu et al., 2015*). Importantly, PG is one of the first enzymes secreted by pathogenic fungi upon contact with plant cell wall and these pectinases have been widely studied for their role in plant pathogenesis especially necrosis and rotting in the infected plants (*De Lorenzo & Ferrari, 2002*; *Kubicek, Starr & Glass, 2014*). This finding corroborated with the reported polygalacturonase activities from the transcriptome profile of *G. boninense*-infected oil palm (*Elaeis guineensis*) whereby PG transcript was shown to be elevated as compared to none observed in control unaffected oil palm (*Ho et al., 2016*). It can be postulated

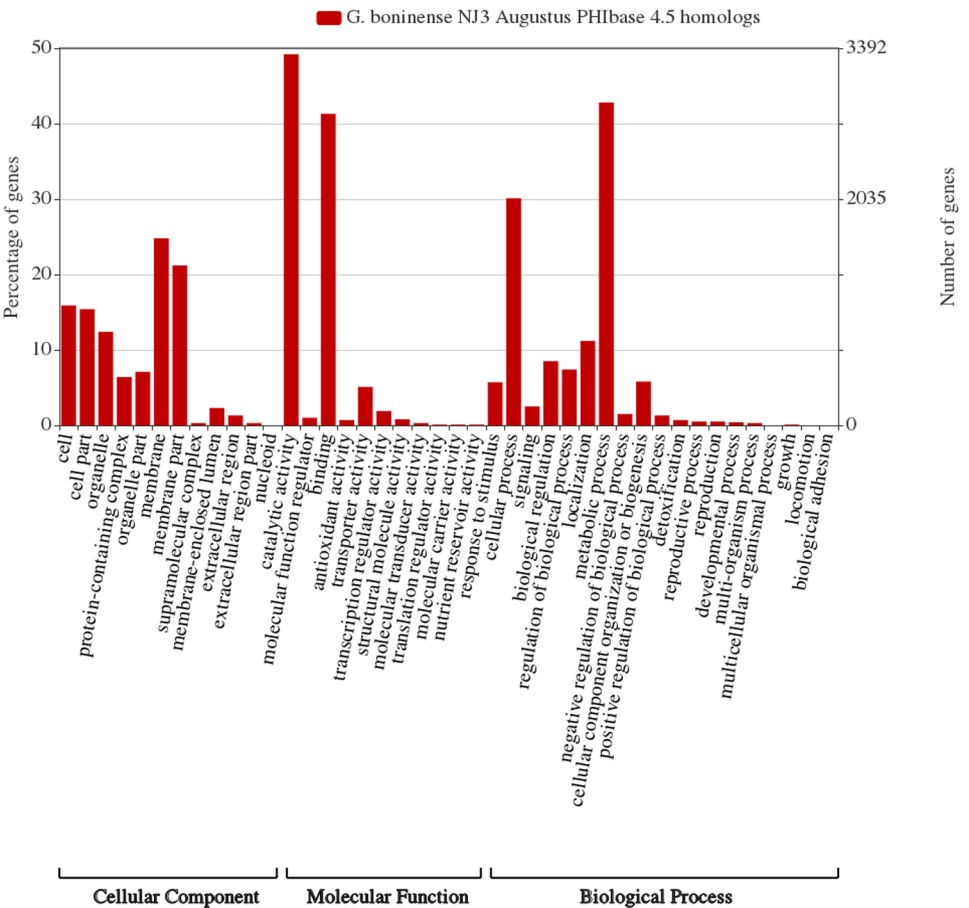

**Figure 4  *G. boninense* NJ3 Augustus predicted amino acid sequences PhiBase 4.5 homologs classified based on gene ontologies.** The gene ontology chart was generated using WEGO 2.0 (http://wego.genomics.org.cn/).

that pectin-acting PG work synergistically with hemicellulases for *Ganoderma* infection in similar fashion to necrotrophic infection and virulence of many phytopathogens including *Fusarium* spp., the main causative agents of vascular wilt and head blight diseases in important crops (*Gómez-Gómez et al., 2002*; *Chen et al., 2012b*; *Paccanaro et al., 2017*). Hemi-cellulosic digestion activities of *G. boninense* were previously demonstrated to assist the fungal growth on oil palm hence supporting the association of these PHI genes as potential pathogenicity factors in oil palm infection (*Surendran et al., 2017*; *Surendran et al., 2018*).

On the other hand, no cutinase (CE5) homolog was found from the CAZymes and PHIbase analysis of *G. boninense* NJ3 suggesting the lack of cutinase-mediated cell wall modification during wood-decaying process which may be compensated by the high numbers of oxidative AAs and hydrolytic GHs found in the *Ganoderma* spp. examined in this study (Table S2). CE5 was found not prevalent in wood-decaying basidiomycetes including pathogenic *H. irregulare* and *Fomitiporia mediterranea* which harbored multiple

**Table 1  List of candidate CWDE-related pathogenicity genes from PHI database analysis.** Information of homolog gene from reference organism is described in the table.

| Gene | PHI ID | Protein ID | Organism | Function/Role | Reference |
|---|---|---|---|---|---|
| PG1 | PHI:4879 | A0A0C4DHY2 | *Fusarium oxysporum* | Catalysis of the random hydrolysis of (1,4)-alpha-D-galactosiduronic linkages in pectate and other galacturonans. | *Bravo Ruiz, Di Pietro & Roncero (2016)* |
| Endo-1,4-beta-xylanase (GH10 family) | PHI:2209 | G4N1Y8 | *Magnaporthe oryzae* | Catalysis of the endohydrolysis of (1,4)-beta-D-xylosidic linkages in xylans | *Nguyen et al. (2011)* |
| Eng1 | PHI:6265 | C0NFK7 | *Histoplasma capsulatum* | Catalysis of the hydrolysis of any O-glycosyl bond | *Garfoot et al. (2016)* |
| LCC2 | PHI:552 | Q96WM9 | *Botrytis cinerea* | Lignin degradation and detoxification of lignin-derived products | *Schouten et al. (2002)* |

copies of lignolytic peroxidase enzymes (*Floudas et al., 2012*; *Zhao et al., 2013*). Expression of cutinase was also reported to be non-essential during pathogenesis of other necrotizing pathogens such as *F. solani* f. sp. *pisi* and *Botrytis cinerea* (*Van Kan et al., 1997*; *Stahl & Schafer, 1992*; *Zhao et al., 2013*). Combined actions of ligninolytic and cellulolytic enzymes including laccase and endoglucanase were previously shown to be directly involved in the wood decaying and infection processes of wheat and cacao by necrotrophic *F. graminearum* and *Moniliophthora roreri,* respectively (*Zhang et al., 2012*; *Meinhardt et al., 2014*). Transcriptome analysis of *Ganoderma* infected-oil palm seedling demonstrated the presence of multiple copies of laccase transcripts as compared to none observed in the sample of beneficial fungus, *T. harzianum*, indicating the important role of cell wall degradation in oil palm infection (*Ho et al., 2016*; *Ho et al., 2018*). The identification of these cell wall degrading PHI genes further supported the hemibiotrophic mode of infection of *G. boninense* conferred by fungal genotypic capabilities to produce a plethora of carbohydrate-acting enzymes. Overall, the comparative genome analysis employed in this study succeeded in characterizing carbohydrate-active enzymes and identifying CWDE genes that are involved in plant cell wall degradation and pathogenesis of *G. boninense*. Further genome analysis of *G. boninense* strains can be carried out with the recent report of draft genome of *G. boninense* G3 strain isolated from Indonesian region (*Utomo et al., 2018*). Correlation between the fungal pathogenicity with CWDE production and other factors can be further validated via targeted transcriptome analysis and gene expression profiling of targeted genes (*Isaac et al., 2018*). Functional studies of the cell wall degrading enzymes in *G. boninense* shall be pursued for greater understanding on the essentiality of the enzymatic capacity in the fungal pathogenesis.

## CONCLUSIONS

In this study, comparative genome analysis succeeded in the identification of carbohydrate-acting and cell wall degrading enzymes in hemibiotrophic *G. boninense* NJ3. The pathogenic *G. boninense* NJ3 genome contained an abundant amount of CAZymes and shared many similar sets of CAZymes to closely-related *G. lucidum* of which the differences between

the gene sets can be attributed to the different nutritional strategy of either *Ganoderma* spp. Necrotizing fungal pathogens including *G. boninense* NJ3 exhibited distinct CAZymes profiles as compared to the non-necrotizing counterparts which can be correlated with host preference and parasitic lifestyles. Several CWDE-related genes were identified from PHI analysis including polygalacturonase and laccase which could directly contribute to the fungal pathogenesis especially through degradation of the plant cell wall. These findings provide fundamental knowledge on the fungal genetic ability and capacity to secrete polysaccharide and cell wall degrading enzymes. Greater insight on the fungal phenotype can be obtained through future studies involving functional and gene expression analysis of specific genes in the fungal carbohydrate metabolism.

## ACKNOWLEDGEMENTS

The authors would like to thank Goh Yit Kheng from Applied Agricultural Resources Sdn Bhd for his insightful revision during the manuscript preparation.

### Funding

This work was supported by Universiti Kebangsaan Malaysia Geran Galakan Penyelidikan-Industri (GGPI-2016-013) and Geran Galakan Penyelidik Muda (GGPM-2017-054) research grants. The article processing fee for this work is funded by DPP-2018-010, a publication initiative grant awarded to the Institute of Systems Biology (INBIOSIS) under UKM Research University Grants. The funders had no role in study design, data collection and analysis, decision to publish, or preparation of the manuscript.

### Grant Disclosures

The following grant information was disclosed by the authors:
Universiti Kebangsaan Malaysia Geran Galakan Penyelidikan-Industri: GGPI-2016-013.
Geran Galakan Penyelidik Muda: GGPM-2017-054.
UKM Research University Grants.

### Competing Interests

The authors declare there are no competing interests.

### Author Contributions

- Ahmad Bazli Ramzi conceived and designed the experiments, performed the experiments, analyzed the data, contributed reagents/materials/analysis tools, prepared figures and/or tables, authored or reviewed drafts of the paper, approved the final draft.
- Muhammad Lutfi Che Me analyzed the data, prepared figures and/or tables, authored or reviewed drafts of the paper, approved the final draft.
- Ummul Syafiqah Ruslan and Syarul Nataqain Baharum analyzed the data, authored or reviewed drafts of the paper, approved the final draft.

- Nor Azlan Nor Muhammad performed the experiments, analyzed the data, contributed reagents/materials/analysis tools, prepared figures and/or tables, authored or reviewed drafts of the paper, approved the final draft.

## Data Availability

The raw data for all CAZymes sequences are available in the Tables S1 and S2.

The Augustus gene annotation is available at Figshare: Ramzi, Ahmad Bazli (2019): Ganoderma boninense NJ3 Augustus gene annotation file. figshare. Journal contribution. Available at DOI: 10.6084/m9.figshare.8969822.v1.

## Supplemental Information

Supplemental information for this article can be found online at http://dx.doi.org/10.7717/peerj.8065#supplemental-information.

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
