# Peer review of "Insight into plant cell wall degradation and pathogenesis of Ganoderma boninense via comparative genome analysis"

_PeerJ, doi:10.7717/peerj.8065_

## Round 0.1 · original submission · Major Revisions

Both reviewers found your work important and solid. However, there are several aspects that can be improved. The reviewers provided a detailed list of their concerns that I urge you to address. For example, it was suggested examine the expression patterns of CWDE genes in G. boninense in comparison with its necrotrophic relatives. It would be also interesting to discuss how incompleteness of genome and gene models affects your results.

Reviewer 1 ·

Basic reporting

This study investigated the CAZymes in G. boninense. Their results provide valuable information for the further studies on the fungal virulence and pathogenicity. Hemibiotrophic fungi have similar CAZyme contents to that of necrotrophic fungi, but obvious distinct infection process. Therefore, it would be of great interest to examine the expression patterns of CWDE genes in G. boninense in comparison with its necrotrophic relatives, which I think should have been done.

Experimental design

no comment

Validity of the findings

no comment

Additional comments

1. The dbCAN release has been updated to the version 6.0 (07/20/2017), why was only the version 1.0 used here?
2. Cutinase (CE5) is an important enzyme family related to the fungal virulence. It is surprised that both G. boninense and G. lucidum have no genes coding cutinase (Table S1). I think authors should re-examine their results carefully to check it.
3. Many conclusions were made in haste in part “Profiling of CAZymes in selected phytopathogenic fungi”, e.g., the authors cannot conclude that GH26 was not observed in other pathogens with necrotrophic behavior (line 294-295) because their comparison was based on only few fungi. Actually, some necrotrophic plant pathogens indeed have GH26 genes.Similarly, authors should conclude those carefully in line 300-301, 325, etc..
Zhao et al. (2013) investigated the distribution of CAZymes among fungi with different nutritional modes, and their results would be helpful for the comparative analysis in this study.
4. Fig.2 B and Fig. 2 C were wrongly referred to in lines 312 and 323.

Reviewer 2 ·

Basic reporting

This paper describes a genomics approach to understanding aspects of the pathogenesis process involved in Ganoderma infection of oil palm plants. The study focused on one aspect of the process, namely the so-called CAZymes involved in cell wall degradation.

Experimental design

the authors have done a good job in carrying out a comparative genome analysis of the CAZyme group of genes in a range of species and pathovars. Their methods are fine and the conclusions are sound.

Validity of the findings

Overall, the study provides useful basic information for the research community.

Additional comments

One cautionary aspect of the study is that so far only a draft version of the G. boninense genome has been published and it is still early days in terms of compiling a fully robust gene model of the species, although such efforts will be helped by the availability of the G. lucidum genome sequence that can be used as a reference genome. As with the oil palm genome, initially sequenced in 2013, it will take some time to come up with a fully reliable gene model and therefore studies such as the present one must be regarded as provisional. It would be useful to make these concerns more apparent by adding a short section in the Introduction.

Despite these reservations, the authors have done a good job in carrying out a comparative genome analysis of the CAZyme group of genes in a range of species and pathovars. Their methods are fine and the conclusions are sound. Overall, the study provides useful basic information for the research community.

---

## Round 0.2 · Minor Revisions

Thank you very much for significantly improving the paper. Analysis of plant pathogens is essential for development of improved agricultural practices. You paper will be of interest for a wide range of plant researchers.

Before acceptance the data generated in this study needs to be made available publicly. Specifically the sequences of the genes predicted from Augustus (or at at least those that were subsequently used in this study) must be deposited in a public repository such as NCBI. A table that relates the GO annotations created in this study to each gene used in this study needs to be provided. Accession numbers or Gene IDs for the genes pulled out from PhyBase need to be provided. The gff file should be provided.

Hope it will not take a significant amount of work to prepare the data files/ Thank you!

Reviewer 1 ·

Basic reporting

The authors have properly addressed all the issues I raised, and I have no new comments.

Experimental design

no comment

Validity of the findings

no comment

Additional comments

no comment

Reviewer 2 ·

Basic reporting

See below

Experimental design

See below

Validity of the findings

See below

Additional comments

Having looked at the paper I am happy that the authors have done all the required amendments and the paper is now acceptable

---

## Round 0.3 · Minor Revisions

Thank you very much for adding the requested information and submitting the sequences to GenBank. Now your paper has a chance to offer an interesting resource to other scientists.

---

## Round 0.4 · accepted · Accept

Thank you very much for depositing the sequences to the public database and providing the link it in the paper.